# A Facile Hydrothermal Synthesis and Resistive Switching Behavior of α-Fe_2_O_3_ Nanowire Arrays

**DOI:** 10.3390/molecules28093835

**Published:** 2023-04-30

**Authors:** Zhiqiang Yu, Jiamin Xu, Baosheng Liu, Zijun Sun, Qingnan Huang, Meilian Ou, Qingcheng Wang, Jinhao Jia, Wenbo Kang, Qingquan Xiao, Tinghong Gao, Quan Xie

**Affiliations:** 1Faculty of Electronic Engineering, Guangxi University of Science and Technology, Liuzhou 545006, China; xjm_165@163.com (J.X.); liubaosheng@gxust.edu.cn (B.L.); sunzijun@gxust.edu.cn (Z.S.); oumlian@mail2.sysu.edu.cn (M.O.);; 2Institute of Advanced Optoelectronic Materials and Technology, College of Big Data and Information Engineering, Guizhou University, Guiyang 550025, Chinaqxie@gzu.edu.cn (Q.X.); 3Wuhan National Laboratory for Optoelectronics, School of Optical and Electronic Information, Huazhong University of Science and Technology, Wuhan 430074, China

**Keywords:** hydrothermal process, α-Fe_2_O_3_ nanowire arrays, memory device, nonvolatile, conducting nanofilaments, oxygen vacancies

## Abstract

A facile hydrothermal process has been developed to synthesize the α-Fe_2_O_3_ nanowire arrays with a preferential growth orientation along the [110] direction. The W/α-Fe_2_O_3_/FTO memory device with the nonvolatile resistive switching behavior has been achieved. The resistance ratio (R_HRS_/R_LRS_) of the W/α-Fe_2_O_3_/FTO memory device exceeds two orders of magnitude, which can be preserved for more than 10^3^s without obvious decline. Furthermore, the carrier transport properties of the W/α-Fe_2_O_3_/FTO memory device are dominated by the Ohmic conduction mechanism in the low resistance state and trap-controlled space-charge-limited current conduction mechanism in the high resistance state, respectively. The partial formation and rupture of conducting nanofilaments modified by the intrinsic oxygen vacancies have been suggested to be responsible for the nonvolatile resistive switching behavior of the W/α-Fe_2_O_3_/FTO memory device. This work suggests that the as-prepared α-Fe_2_O_3_ nanowire-based W/α-Fe_2_O_3_/FTO memory device may be a potential candidate for applications in the next-generation nonvolatile memory devices.

## 1. Introduction

As the fourth fundamental element, the resistive switching random access memory (ReRAM) with the merits of fast memory speed, good endurance, high integration density, long retention, ultra-low power dissipation, high scalability, and multilevel properties [1,2,3,4,5,6,7,8,9,10,11,12,13,14,15,16,17,18,19,20,21,22,23,24,25,26,27,28,29,30,31,32,33] has been regarded as one of the most outstanding candidates for next-generation nonvolatile memory applications, which defines the reversal resistive switching between the high resistance state (HRS) and the low resistance state (LRS) as the logic “0” and “1”. In the last decade, various organic and inorganic nanomaterials have been used for ReRAM applications owing to their unique physical and chemical features [2]. Among them, the transition metal oxide nanomaterials, including Fe_2_O_3_ [3,4,5,6,7,8,9,10,11,12,13,14,15,16,17,18,19,20,21], ZnO [22], TiO_2_ [23], HfO_2_ [24], GaO_x_ [25], Co_3_O_4_ [26], CuO [27], WO_x_ [28], NiO [29], In_2_O_3_ [30], TaO_x_ [31], and CeO_2_ [32] have received wide attention because of their facile fabrication process, good compatibility, simple composition as well as excellent resistive switching properties. In recent decades, increasing interest has been devoted to the hematite α-Fe_2_O_3_ nanomaterials due to their advantages in terms of the high chemical stability, nontoxicity, low cost, and appropriate band gap [33,34,35] (2.2 eV). The hematite α-Fe_2_O_3_ nanomaterials have shown promising potential for future nonvolatile memory applications [3,8,10,11,13].

The hematite α-Fe_2_O_3_ is an n-type semiconductor with a corundum-type structure and weakly ferromagnetic above the Morin temperature (T_M_ ≈ 260 K). In comparison with the other iron oxides, such as magnetite Fe_3_O_4_ and maghemite γ-Fe_2_O_3_, the hematite α-Fe_2_O_3_ is the most stable iron oxide under ambient conditions and frequently occurring polymorph with a rhombohedral-hexagonal structure [8,33]. In particular, the hematite α-Fe_2_O_3_ provides more advantages over the other iron oxides due to its greater chemical inertness, low toxicity, and also owing to its high natural abundance [34,35]. Furthermore, the hematite α-Fe_2_O_3_ possesses an appropriate band gap value between 1.9 and 2.2 eV [33,34,35], thus permitting a strong visible-light absorption which transforms to be the maximal theoretical solar-to-hydrogen (STH) efficiency (16.8%) in the visible region. Though, the hematite α-Fe_2_O_3_ holds a CB position with considerably more positive proton reduction potential and has the capability to be utilized for PEC water oxidation with an external bias. Additionally, the hematite α-Fe_2_O_3_ has some shortcomings, such as the short hole diffusion length (L_D_ < 5 nm), minor carrier lifetime (Lifetime < 10 ps), poor light absorption near the band edge, sluggish minority charge carrier (hole) mobility, comparatively small optical absorption coefficient, low electrical conductivity, and the incompatibility of the band edge for water redox reactions [35]. To overcome these challenges, one effective approach could be controlling the morphology, porosity, and size of the hematite α-Fe_2_O_3_ nanostructures. In recent years, the hematite α-Fe_2_O_3_ nanomaterials with the unique shape and size have received increasing attention due to their peculiar and fascinating physicochemical properties, superior long-term stability, stability in aqueous alkaline solutions, low processing cost, environmental inertness, and high resistance to corrosion, which make the hematite α-Fe_2_O_3_ an excellent multifunctional material for applications in gas sensors, pigments, magnetic materials, photocatalysts, photoelectrochemical (PEC) electrodes, drug delivery, and tissue repairing engineering, nanoelectronic and optoelectronic devices [3,4,5,6,7,8,9,10,11,12,13,14,15,16,17,18,19,20,21,22,23,24,25,26,27,28,29,30,31,32,33,34,35,36,37,38].

In the last decades, various Fe_2_O_3_-based ReRAMs, such as the Ag/Fe_2_O_3_/Ti [3], Pt/Fe_2_O_3_/Pt/Ti [4], Ag/BaTiO_3_/γ-Fe_2_O_3_/ZnO/Ag [5], Au/Pt-Fe_2_O_3_/Ti [6], Au/Fe_2_O_3_/FTO [7,8], Ag/Fe_2_O_3_/ZnO/ITO [9], Ag/Fe_2_O_3_/FTO [10], Ag/TiO_2_/Fe_2_O_3_/FTO [11], Ag/Fe_2_O_3_-PVA/FTO [12], Ag/BiFeO_3_/γ-Fe_2_O_3_/FTO [13], top-probe/α-Fe_2_O_3_/ZnO/bottom-probe [14], W/Fe_2_O_3_ NC film/Pt [15], Ti/γ-Fe_2_O_3_-NPs/Pt [16], Ti/Pt-Fe_2_O_3_ core-shell NPs/Pt/PES [17], Ti/Pt-Fe_2_O_3_-core-shell-nanoparticles/γ-Fe_2_O_3_-nanoparticles/Pt [18], Ti/Fe_2_O_3_-SiO_2_/Si [19], Cr/ZnO/Pt-Fe_2_O_3_ NPs/ZnO/Cr [20], and Au/HfSiO/γ-Fe_2_O_3_/Ni_2_O_3_/HfSiO/Pt [21] have shown the excellent resistive switching properties for the nonvolatile memory applications. Furthermore, several resistive switching mechanisms corresponding to the formation and rupture of conducting filaments [4,15,16,18,20], the space-charge-limited conduction [6,9,17], the Schottky emission [8,11], and the valence change mechanisms [12] have been developed to illustrate the resistive switching behaviors of the Fe_2_O_3_ nanomaterials-based memory devices. Recently, many preparation processes, including the magnetron sputtering technique [4,5], the dip coating method [6,15,16,20], the spin coating technique [3,7,9,14,21], the ultrasonic spray pyrolysis [8], the hydrothermal method [10,11,17,18], the co-precipitation method [12,13], and the atomic layer deposition process [19] have been carried out to prepare the Fe_2_O_3_-based memory devices. Among them, the hydrothermal method is suitable for the controlled synthesis of large-scale α-Fe_2_O_3_ nanomaterials owing to its low cost, simple process, and relatively mild reaction conditions, which is one of the effective methods for preparing the α-Fe_2_O_3_ nanowire arrays. However, there have been no such reports about the synthesis and resistive switching behaviors of the α-Fe_2_O_3_ nanowire-based memory devices so far. Moreover, a proper theory to explain the nonvolatile resistive switching mechanism of the α-Fe_2_O_3_ nanowire-based W/α-Fe_2_O_3_/FTO memory device is still urgently desired.

In this work, a facile hydrothermal process was performed to synthesize the hematite α-Fe_2_O_3_ nanowire arrays with a preferential growth orientation along the [110] direction on the FTO substrates. The nonvolatile bipolar resistive switching behavior of the W/α-Fe_2_O_3_/FTO memory device has been achieved. The partial formation and breakup of conducting nanofilaments modified by the intrinsic oxygen vacancies in the as-preparedα-Fe_2_O_3_ nanowire arrays have been suggested to be responsible for the nonvolatile resistive switching behavior of the W/α-Fe_2_O_3_/FTO memory device. This work suggests that the as-prepared α-Fe_2_O_3_ nanowire-based W/α-Fe_2_O_3_/FTO memory device may be a promising candidate for future nonvolatile memory applications.

## 2. Results and Discussion

Figure 1a exhibits the XRD patterns of the as-prepared α-Fe_2_O_3_/FTO, FTO, and the standard diffraction peaks of α-Fe_2_O_3_ (PDF# 33-0664, JCPDS), respectively. It is clear that all the XRD diffraction peaks that appeared upon the α-Fe_2_O_3_ nanowire arrays are indexed to the rhombohedral phase α-Fe_2_O_3_ (PDF# 33-0664, JCPDS) [33,37]. The presence of the strong peaks suggests the high crystalline quality of the as-prepared α-Fe_2_O_3_ nanowire arrays. In particular, the (110) diffraction peak located at 35.2° is significantly enhanced, which implies that the as-prepared α-Fe_2_O_3_ nanowire arrays are highly oriented with a preferential growth orientation along the [110] direction. Figure 1b displays the top-view FESEM image of the as-prepared α-Fe_2_O_3_ nanowire arrays. As shown in Figure 1b, the size distribution on the entire surface of the α-Fe_2_O_3_ nanowire arrays is relatively uniform. Figure 1c shows the cross-sectional FESEM image of the as-prepared α-Fe_2_O_3_ nanowire arrays. It can be found that the α-Fe_2_O_3_ nanowire arrays with a length of about 550 nm have been observed, which can be acted as the dielectric material layer of the W/α-Fe_2_O_3_/FTO memory device. In addition, the diameter distribution histogram of the as-prepared α-Fe_2_O_3_ nanowire arrays further indicates that the average diameter of the α-Fe_2_O_3_ nanowire arrays is approximately 37.5 nm, as revealed in Figure 1d.

The average crystalline size *D* of the as-prepared α-Fe_2_O_3_ nanowire arrays can be given by Scherer’s equation [36] as follows:(1)$$D=\frac{0.9\lambda }{\beta cos\theta }\;$$
where *β* is fullwidth at half-maxima, *λ* is the X-ray wavelength (1.5406 Å), and *θ* is the diffraction angle. The average crystalline size of the as-prepared α-Fe_2_O_3_ nanowire arrays is calculated to be 35 nm, suggesting the nanocrystalline nature of the as-prepared α-Fe_2_O_3_ nanowire arrays.

Figure 2 indicates the survey XPS spectra, the Fe 2p and O 1s high-resolution XPS spectra of the as-prepared α-Fe_2_O_3_ nanowire arrays. It is found that all the Fe and O elements, together with the C element, can be observed in the as-prepared α-Fe_2_O_3_ nanowire arrays, where the C element comes from the carbon source in air adsorbed on the surface of the as-prepared α-Fe_2_O_3_ nanowire arrays, which is employed to calibrate the other elements including the Fe and O elements, as shown in Figure 2a. Figure 2b shows the Fe 2p high-resolution XPS spectrum of the as-prepared α-Fe_2_O_3_ nanowire arrays. The Fe 2p XPS peaks located at 724.14 eV and 711.70 eV are attributed to the Fe 2p_1/2_ and Fe 2p_3/2_, respectively. The Fe 2p XPS peak at 709.92 eV is associated with Fe^2+^ [37]. Furthermore, two distinguishable satellite peaks that appeared at 717.47 eV and 732.70 eV can be observed, which implies the presence of Fe^3+^ in the α-Fe_2_O_3_ nanowire arrays [15,34,38]. As revealed in Figure 2c, the O 1s high-resolution XPS spectrum can be decomposed into three Gaussian fitting peaks corresponding to 529.53 eV, 530.85 eV, and 532.57 eV, which are assigned to the lattice oxygen, oxygen vacancies, and the chemisorbed oxygen species [14,33,38], respectively. In particular, the relative concentration of oxygen vacancies can be calculated to be 33.8% from the peak area, which is larger than that of the chemisorbed oxygen species (23.6%) for the as-prepared α-Fe_2_O_3_ nanowire arrays. The fitting results of the XPS spectra suggest that a considerable amount of oxygen vacancies exist in the as-prepared α-Fe_2_O_3_ nanowire arrays, which can act as the trapping center and be responsible for the nonvolatile resistive switching behavior of the as-prepared α-Fe_2_O_3_ nanowire-based W/α-Fe_2_O_3_/FTO memory device.

Figure 3a and Figure 3b indicate the UV-Visible absorption spectra and the corresponding Tauc plots of the as-prepared α-Fe_2_O_3_ nanowire arrays, respectively. As shown in Figure 3a, the as-prepared α-Fe_2_O_3_ nanowire arrays display excellent optical absorption capacity with an absorption edge of about 645 nm in the visible absorption region. In addition, the optical band gap (*Eg*) of the as-prepared α-Fe_2_O_3_ nanowire arrays can be evaluated with the following Tauc relation [3,38]:(2)$${\left(\alpha h\upsilon \right)}^{k}=A\left(h\upsilon -E_{g}\right)$$
where *α* is the absorbance coefficient of the as-prepared α-Fe_2_O_3_ nanowire arrays, *h* is Planck’s constant, *v* is the vibration frequency, and *A* is the optical constant. Moreover, *k* is the vibration frequency of α-Fe_2_O_3_ (k=0.5) due to its indirect band gap. As depicted in Figure 3b, the optical band gap of the α-Fe_2_O_3_ nanowire arrays can be found to be 1.92 eV, which is lower than that of the pristine α-Fe_2_O_3_ [33,34,35] (2.2 eV), further confirming the presence of the intrinsic oxygen vacancies in the α-Fe_2_O_3_ nanowire arrays synthesized by the hydrothermal process.

Figure 4a reveals the schematic configuration of the as-prepared α-Fe_2_O_3_ nanowire-based W/α-Fe_2_O_3_/FTO memory device, composed of the top W electrode, the α-Fe_2_O_3_ nanowire arrays, and the bottom FTO electrode. During the tests, the bias voltages of the W/α-Fe_2_O_3_/FTO memory device were applied to the top W electrode with the bottom FTO electrode grounded. The typical *I-V* curve plotted in a semi-logarithmic scale has been obtained by sweeping the applied voltage in the range of 0 V → +3 V → 0 V → −4 V → 0 V with a compliance current of 100 mA to protect the W/α-Fe_2_O_3_/FTO memory device from the unrecoverable breakdown, as shown in Figure 4b. The arrows and numbers, as recorded in Figure 4b, indicate the applied voltage sweeping direction and sequence in the device. It is clearly shown that the W/α-Fe_2_O_3_/FTO memory device demonstrates stable nonvolatile bipolar resistive switching behavior. The pristine resistance state of the W/α-Fe_2_O_3_/FTO memory device is the high resistance state (HRS). When the applied voltage rises from 0 V to +3 V, the W/α-Fe_2_O_3_/FTO memory device will vary from the HRS to LRS with a sudden current jump at +0.98 V (V_set_), which means that the set process occurs. After that, the W/α-Fe_2_O_3_/FTO memory device will maintain the LRS until the applied voltage reduces to −3.11 V (V_reset_), thus demonstrating the nonvolatile resistive switching behavior of the device. Subsequently, the reset process occurs at V_reset_, which induces a recovery to the pristine HRS with a drastic drop of current in the device, indicating the nonvolatile bipolar resistive switching behavior of the W/α-Fe_2_O_3_/FTO memory device.

In order to evaluate the nonvolatile resistive switching behavior of the as-prepared α-Fe_2_O_3_ nanowire-based W/α-Fe_2_O_3_/FTO memory device, Figure 4c exhibits the resistance-voltage characteristic of the device under the voltage sweep of 0 V → +3 V → 0 V → −4 V → 0 V. It is obvious that a highly stable nonvolatile resistive switching window between the HRS and LRS can be observed clearly, which suggests the reliable resistive switching behavior of the W/α-Fe_2_O_3_/FTO memory device. Figure 4d indicates the retention capability of the W/α-Fe_2_O_3_/FTO memory device at the reading voltage of 0.1 V. Significantly, the resistance ratio (R_HRS_/R_LRS_) between the HRS and LRS exceeds two orders of magnitude, which can be stably preserved for over 10^3^ s without obvious decline. In comparison with the previous literatures on the α-Fe_2_O_3_-based memory devices, as tabulated in Table 1 [3,4,5,6,7,8,9,10,11,12,13,14,15,16,17,18,19,20,21], the as-prepared W/α-Fe_2_O_3_/FTO memory device in this work indicates a relatively lower V_set_ (+0.98 V) and an appropriate resistance ratio of more than two orders of magnitude, which demonstrates the promising potential of the as-prepared α-Fe_2_O_3_ nanowire-based W/α-Fe_2_O_3_/FTO memory device for applications in future nonvolatile memory devices.

To further illustrate the resistive switching mechanism of the as-prepared α-Fe_2_O_3_ nanowire-based W/α-Fe_2_O_3_/FTO memory device, the *I-V* curves of the device have been plotted in a double-logarithmic scale, as indicated in Figure 5. Figure 5a exhibits the double-logarithmic *I-V* curve of the W/α-Fe_2_O_3_/FTO memory device in the positive voltage region. In the LRS region, the linear *I-V* curve with a slope of 1.09 indicates the device’s Ohmic conduction mechanism (*I∝V*). In the HRS region, the slope of the *I-V* curve varies from 1.02 to 1.79 and then to 2.36 with the rise of the positive voltage. The complicated *I-V* curve in the HRS can be deconvoluted into three distinct regions corresponding to the Ohmic region (*I∝V*), the Child’s law region (*I*∝*V*^2^), and the trap-filled limited region (*I*∝*V*^m^, m > 2), respectively, suggesting the trap-controlled space-charge-limited current (SCLC) mechanism [8] of the device in the HRS. In the low voltage region, the thermally generated carriers dominate the conduction behavior, and the device shows the linear Ohmic conduction mechanism in the HRS. With the increase of the positive voltage, the injected carriers gradually become predominant and occupy the trapping center composed of oxygen vacancies, and the resistive switching behavior of the W/α-Fe_2_O_3_/FTO memory device would follow the Child’s law and the trap-filled limited conduction mechanisms in turn. After then, the set process occurs at V_set,_ and the device would switch to the LRS with a rapid increase in current, implying the formation of conducting nanofilaments. Figure 5b indicates the double-logarithmic *I-V* curve of the W/α-Fe_2_O_3_/FTO memory device in the negative voltage region. A similar resistive switching mechanism of the device can be found in the negative voltage region compared with that in the positive voltage region. The linear *I-V* curve with a slope of 0.98 reveals the Ohmic conduction mechanism of the device in the LRS region. With the increase of the negative voltage, the device will remain the LRS till a large enough negative sweeping voltage V_reset_ is applied, indicating the nonvolatile resistive switching behavior of the W/α-Fe_2_O_3_/FTO memory device. After that, the reset process occurs at V_reset._ The device would transition from the LRS to the HRS with a dramatic decrease of current, which suggests the rupture of conducting nanofilaments. Subsequently, the slope of the *I-V* curve changes from 5.49 to 2.15 and then to 1.19 with the drop of the negative voltage in the HRS region, which also demonstrates the trap-controlled space-charge-limited current mechanism of the as-prepared W/α-Fe_2_O_3_/FTO memory device in the HRS region.

In the Ohmic region, the current density $J_{o}$ can be expressed as
(3)$$\;\;J_{o}=qn\mu \frac{V\;}{d}$$
where *q* is the elementary charge, *n* is the thermally generated free carrier density, *μ* is the electron mobility, *V* is the applied voltage, and *d* is the thickness of α-Fe_2_O_3_ oxide layer in the W/α-Fe_2_O_3_/FTO memory device.

In the Child’s law region, the current density $J_{c}$ can be given as
(4)$$\;\;J_{c}=\frac{9}{8}\varepsilon \mu \theta \frac{V^{2}\;}{d^{3}}$$
where *ε* is the dielectric constant of the α-Fe_2_O_3_ oxide layer, and *θ* is the proportion of the free carrier density to the total carrier density.

The as-prepared α-Fe_2_O_3_ nanowire-based W/α-Fe_2_O_3_/FTO memory device switches from the HRS to LRS in the set process and then to HRS in the reset process, which might be consistent with the partial formation and rupture of conducting nanofilaments, respectively. Thus, the partial formation and breakup of conducting nanofilaments modified by the intrinsic oxygen vacancies in the as-prepared α-Fe_2_O_3_ nanowire arrays have been suggested to be responsible for the nonvolatile bipolar resistive switching behavior of the α-Fe_2_O_3_ nanowire-based W/α-Fe_2_O_3_/FTO memory device.

As mentioned above, the nonvolatile resistive switching behavior of the as-prepared α-Fe_2_O_3_ nanowire-based W/α-Fe_2_O_3_/FTO memory device might be attributed to the partial formation and breakup of conducting nanofilaments modified by the intrinsic oxygen vacancies in the as-prepared α-Fe_2_O_3_ nanowire arrays. In the set process, as shown in Figure 6, when the positive sweeping voltage is applied to the W/α-Fe_2_O_3_/FTO memory device in the HRS, the oxygen ions will drift upward and accumulate at the top W electrode with the increase of the positive voltage. In contrast, the oxygen vacancies migrate from the top W electrode to the bottom FTO electrode, setting up the metallic conducting nanofilaments across the α-Fe_2_O_3_ nanowire arrays. Once the partial formation of conducting nanofilament occurs at V_set_ with the increasing applied voltage, the electrons are injected from the bottom FTO electrode to the top W electrode along the conducting nanofilament, which would induce a switch from the HRS to LRS with an abrupt jump of current in the W/α-Fe_2_O_3_/FTO memory device. Subsequently, the device will keep the LRS until a large enough negative sweeping voltage V_reset_ is applied, indicating the nonvolatile resistive switching behavior of the device. In the reset process, when the negative sweeping voltage is applied to the W/α-Fe_2_O_3_/FTO memory device, the oxygen vacancies will move toward the top W electrode and recombine with oxygen ions at the W/Fe_2_O_3_ interface, resulting in the partial rupture of conductive nanofilaments. Subsequently, the device switches to the pristine HRS with a dramatic current decline at V_reset_. Thus, the partial formation and breakup of conducting nanofilaments modified by the intrinsic oxygen vacancies in the as-prepared α-Fe_2_O_3_ nanowire arrays have been suggested to be responsible for the nonvolatile resistive switching behavior of the W/α-Fe_2_O_3_/FTO memory device. This work suggests that the as-prepared α-Fe_2_O_3_ nanowire-based W/α-Fe_2_O_3_/FTO memory device may be a potential candidate for future nonvolatile memory applications.

## 3. Experimental Section

All the reagents, including iron chloride hexahydrate (FeCl_3_·6H_2_O, 99%), isopropyl alcohol ((CH_3_)_2_CHOH, 99.7%), and acetone (CH_3_COCH_3_, 99.8%), provided by Sigma-Aldrich, are of analytical grade and used directly without further purifying. The fluorine-doped tin oxide (FTO, 15 Ω/square) substrates are purchased from the NSG Pilkington and rinsed ultrasonically in acetone, isopropyl alcohol, and deionized water, and then dried at room temperature.

The α-Fe_2_O_3_ nanowire arrays were synthesized by a facile hydrothermal process. Initially, 5 mM of FeCl_3_·6H_2_O was dissolved into the 15 mL of deionized water. After vigorous stirring at 30 °C for 10 min, the yellow transparent solution was generated, acting as the α-Fe_2_O_3_ nanowire arrays precursor solution. The precursor solution was transferred into a Teflon-lined stainless-steel autoclave. A piece of cleaned FTO substrate with the conducting surface facing down was placed at an angle against the inner wall of the Teflon-liner. Then, the stainless-steel autoclave was annealed in a muffle furnace at 100 °C for 6 h. Subsequently, the as-prepared sample was taken out, cleared thoroughly with deionized water, and dried at room temperature. After calcination at 800 °C for 20 min, the α-Fe_2_O_3_ nanowire arrays were achieved. The circular top W electrodes with a diameter of 10 μm in the as-prepared α-Fe_2_O_3_ nanowire-based W/α-Fe_2_O_3_/FTO memory device were directly deposited on the α-Fe_2_O_3_ nanowire arrays by the magnetron sputtering process with a metal shadow mask.

The crystalline phases and morphologies of the hematite α-Fe_2_O_3_ nanowire arrays were tested by X-ray diffraction (XRD, PANalytical PW3040/60) with Cu Kα radiation (λ = 0.1541 nm) and Field-emission scanning electron microscopy (FESEM, FEI Nova NanoSEM 450), respectively. The chemical states of the hematite α-Fe_2_O_3_ nanowire arrays were identified by the X-ray photoelectron spectroscopy (XPS, AXIS-ULTRA DLD-600W) with monochromatic Al Kα radiation (hv = 1486.6 eV). The current-voltage (*I-V*) characteristics of the W/α-Fe_2_O_3_/FTO memory device were measured by an Agilent B2901A semiconductor parameter analyzer. During the measurements, the bias voltages of the as-prepared α-Fe_2_O_3_ nanowire-based W/α-Fe_2_O_3_/FTO memory device were applied to the top W electrode, and the bottom FTO electrode was grounded. Compliance current fixed at 100 mA was adopted to protect the device from the unrecoverable breakdown. All the above processes were operated under atmospheric conditions.

## 4. Conclusions

In summary, the rhombohedral phase α-Fe_2_O_3_ nanowire arrays with the preferential growth orientation along the [110] direction and an average diameter of approximately 37.5 nm have been fabricated by a facile hydrothermal method. The α-Fe_2_O_3_ nanowire-based W/α-Fe_2_O_3_/FTO memory device has been prepared for the first time. The as-prepared α-Fe_2_O_3_ nanowire-based W/α-Fe_2_O_3_/FTO memory device indicates the nonvolatile bipolar resistive switching behavior with a relatively lower V_set_ (+0.98 V) and an appropriate resistance ratio of more than two orders of magnitude, which can be preserved for more than 10^3^ s without obvious deterioration. Furthermore, the carrier transport properties of the as-prepared W/α-Fe_2_O_3_/FTO memory device are associated with the Ohmic conduction mechanism in the low resistance state and the trap-controlled space-charge-limited current conduction mechanism in the high resistance state, respectively. In addition, the partial formation and breakup of conducting nanofilaments modified by the intrinsic oxygen vacancies in the as-prepared α-Fe_2_O_3_ nanowire arrays have been suggested to be responsible for the nonvolatile bipolar resistive switching behavior of the W/α-Fe_2_O_3_/FTO memory device. This work demonstrates that the as-prepared α-Fe_2_O_3_ nanowire-based W/α-Fe_2_O_3_/FTO memory device may be a promising candidate for applications in future nonvolatile memory devices.

## Figures and Tables

**Figure 1 molecules-28-03835-f001:**
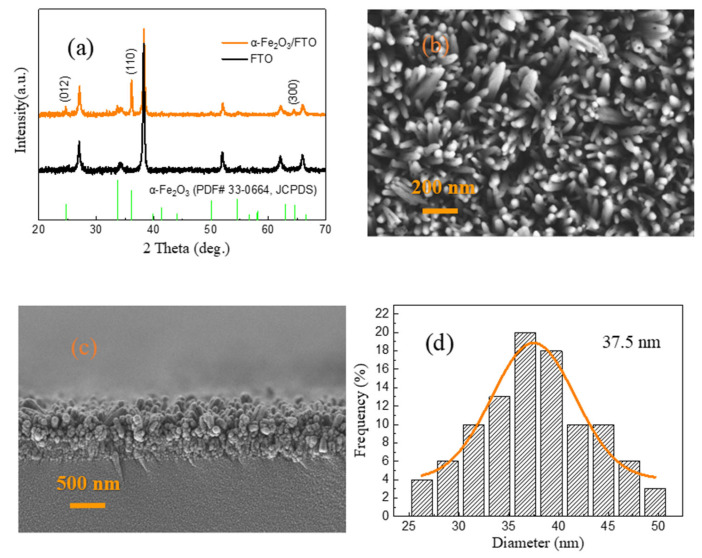
(**a**) XRD patterns of the as-prepared α-Fe_2_O_3_/FTO, FTO, and the standard diffraction peaks of α-Fe_2_O_3_ (PDF# 33-0664, JCPDS), respectively. (**b**) Top-view FESEM image of the as-prepared α-Fe_2_O_3_ nanowire arrays. (**c**) Cross-sectional FESEM image of the as-prepared α-Fe_2_O_3_ nanowire arrays. (**d**) Diameter distribution histogram of the as-prepared α-Fe_2_O_3_ nanowire arrays.

**Figure 2 molecules-28-03835-f002:**
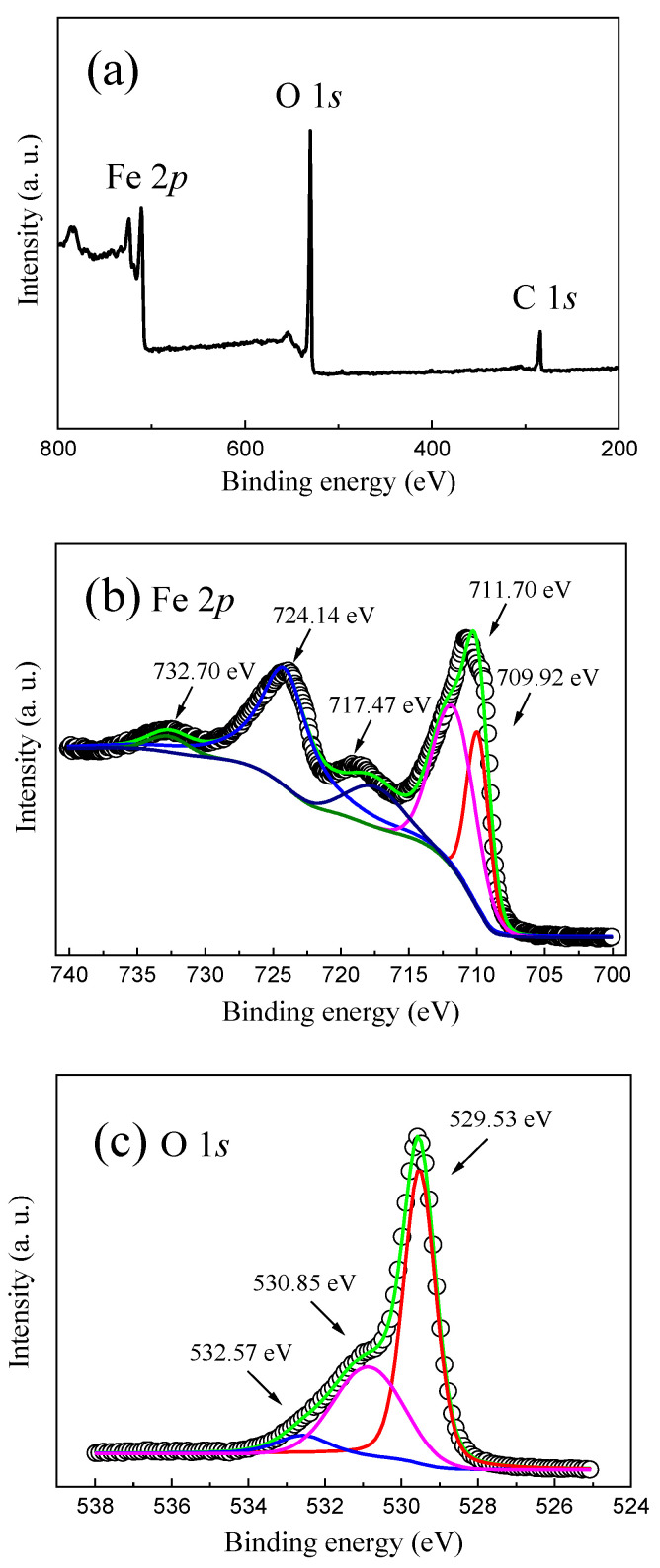
(**a**) Survey XPS spectra of the as-prepared α-Fe_2_O_3_ nanowire arrays. (**b**) Fe 2*p* and (**c**) O 1*s* high-resolution XPS spectra of the as-prepared α-Fe_2_O_3_ nanowire arrays.

**Figure 3 molecules-28-03835-f003:**
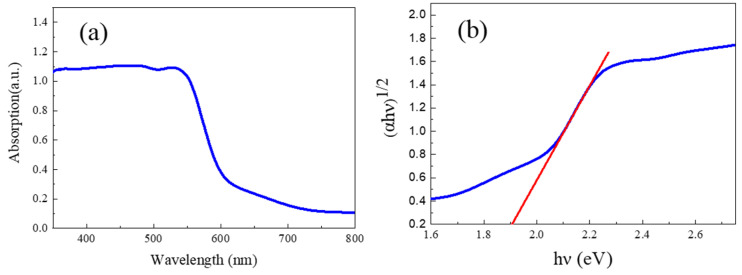
(**a**) UV-Visible absorption spectra of the α-Fe_2_O_3_ nanowire arrays. (**b**) Tauc plots of the α-Fe_2_O_3_ nanowire arrays.

**Figure 4 molecules-28-03835-f004:**
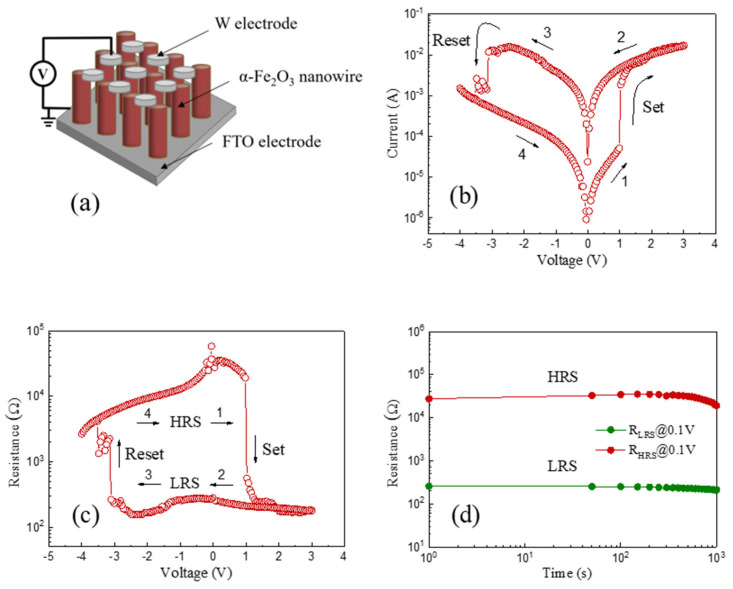
(**a**) The schematic configuration of the as-prepared α-Fe_2_O_3_ nanowire-based W/α-Fe_2_O_3_/FTO memory device. (**b**) The semi-logarithmic *I-V* curve of the device. (**c**) Resistance-voltage characteristic of the device. (**d**) Retention capability of the device.

**Figure 5 molecules-28-03835-f005:**
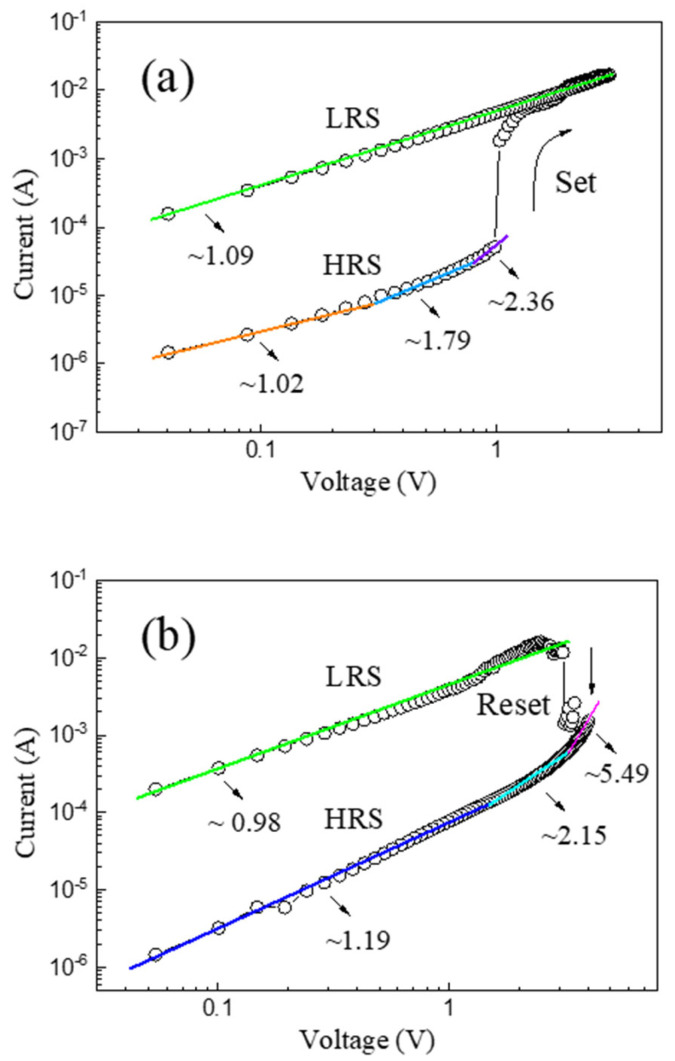
The double-logarithmic *I-V* curves of the as-prepared α-Fe_2_O_3_ nanowire-based W/α-Fe_2_O_3_/FTO memory device in the (**a**) positive voltage region and (**b**) negative voltage region.

**Figure 6 molecules-28-03835-f006:**
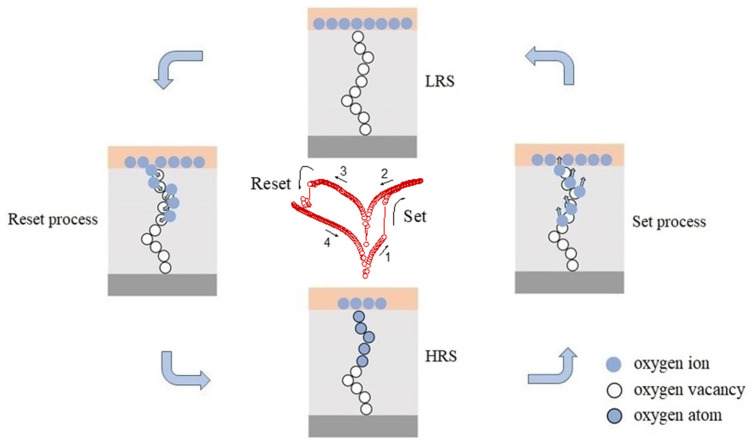
Schematic of the nonvolatile resistive switching mechanism of the as-prepared α-Fe_2_O_3_ nanowire-based W/α-Fe_2_O_3_/FTO memory device.

**Table 1 molecules-28-03835-t001:** Comparison of the performance parameters of the α-Fe_2_O_3_-based memory devices.

Device Structure	V_set_/V_reset_ (V)	Preparation Method	R_HRS_/R_LRS_ Ratio	Retention	Reference
Ag/Fe_2_O_3_/Ti	~+0.02/−0.1	Spin coating technique	~10	3.6 × 10^3^ s	[3]
Pt/Fe_2_O_3_/Pt/Ti	+1.4/+0.5	Magnetron sputtering technique	~7	-	[4]
Ag/[BiFeO_3_/γ-Fe_2_O_3_]/FTO	+0.98/−1.38	Magnetron sputtering technique	~10	-	[5]
Au/Pt-Fe_2_O_3_/Ti	~−1/~+3.4	Dip coating method	~10	5 × 10^4^ s	[6]
Ag/γ-Fe_2_O_3_ films/FTO	+1.85/−1.25	Spin coating technique	-	-	[7]
Au/Fe_2_O_3_/FTO	~+1.5/~−1.2	Ultrasonic spray pyrolysis	~10	10 h	[8]
Ag/Fe_2_O_3_/ZnO/ITO	+0.9/−1	Spin coating technique	~90.1	30 days	[9]
Ag/Fe_2_O_3_/FTO	~+2/~−2	Hydrothermal method	~10^4^	-	[10]
Ag/[TiO_2_/α-Fe_2_O_3_]/FTO	~+4/~−4	Hydrothermal method	~10	10^3^ s	[11]
Ag/Fe_2_O_3_-PVA/FTO	+2/−0.7	Co-precipitation method	~10	-	[12]
Ag/BaTiO_3_/γ-Fe_2_O_3_/ZnO/Ag	+3.1/−4.7	Co-precipitation method	~10	-	[13]
top-probe/α-Fe_2_O_3_/ZnO/bottom-probe	−0.55/-	Spin coating technique	~20	10^3^ s	[14]
W/Fe_2_O_3_ NC film/Pt	~−1.2/~+1.6	Dip coating method	>10^2^	10^5^ s	[15]
Ti/γ-Fe_2_O_3_-NPs/Pt	+1~+2/−1	Dip coating method	~10^2^	-	[16]
Ti/Pt-Fe_2_O_3_ core-shell NPs/Pt/PES	+2.5/-	Hydrothermal method	<10	-	[17]
Ti/Pt-Fe_2_O_3_-core-shell/γ-Fe_2_O_3_/Pt	~+1/~−1	Hydrothermal method	~10^2^	-	[18]
Ti/Fe_2_O_3_-SiO_2_/Si	+2/−2.5	Atomic layer deposition process	2.8	-	[19]
Cr/ZnO/Pt-Fe_2_O_3_ NPs/ZnO/Cr	−7/+7	Dip coating method	~5	10^4^ s	[20]
Au/HfSiO/γ-Fe_2_O_3_/Ni_2_O_3_/HfSiO/Pt	+1.96/−1.90	Spin coating technique	~10^2^	-	[21]
W/α-Fe_2_O_3_/FTO	+0.98/−3.11	Hydrothermal method	>10^2^	>10^3^ s	This work

## Data Availability

The data presented in this study are available on request from the corresponding author.

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
