# Peer review of "A Facile Hydrothermal Synthesis and Resistive Switching Behavior of α-Fe2O3 Nanowire Arrays"

_molecules, 2023, doi:10.3390/molecules28093835_

Round 1

Reviewer 1 Report

1.     It is not clear what the authors mean by “facile hydrothermal synthesis”? Hydrothermal synthesis is quite a complicated method; however, it brings many benefits.

2.     Authors did not develop a hydrothermal process but applied it for the synthesis. What is more important, is that the authors used a combined method of hydrothermal synthesis and thermal treatment.

3.     In the introduction it is necessary to describe the possible synthesis methods and explain their advantages and disadvantages. At the moment it is not clear if this is the first time when hydrothermal synthesis was applied to obtain Fe2O3.

4.     Abstract and conclusions are more or less the same. Authors should overwrite them according to requirements for abstracts and conclusions.

5.     Figure 1 is not described in the text.

6.     Please full file XRD spectra by etalon spectra of No 33-0664.

7.     Probably authors used PDF-2 or PDF-4 database but not JCPDS?

Author Response

Thank you very much for your letter and for the reviewer’s comments concerning our manuscript entitled “A facile hydrothermal synthesis and resistive switching behavior of α-Fe2O3 nanowire arrays”. (Manuscript ID: molecules-2337904).

Those comments are all valuable and very helpful for revising and improving our paper, as well as the important guiding significance to our researches. The manuscript has been carefully revised and the major revisions were marked in red. The point-by-point answers to the comments and suggestions were listed as below.

Responds to the reviewer’s comments:

  1. It is not clear what the authors mean by “facile hydrothermal synthesis”? Hydrothermal synthesis is quite a complicated method; however, it brings many benefits.

Response: Thank you very much for your comments and suggestions.

In this paper, a facile hydrothermal process has been developed to synthesize the α-Fe2O3 nanowire arrays on the FTO substrate. The hydrothermal method is suitable for the controlled synthesis of the large-scale α-Fe2O3 nanomaterials owing to its low cost, simple process, relatively mild reaction conditions, which is one of the effective methods to prepare the α-Fe2O3 nanowire arrays.

  1. Authors did not develop a hydrothermal process but applied it for the synthesis. What is more important, is that the authors used a combined method of hydrothermal synthesis and thermal treatment.

Response: Thank you very much for your comments and suggestions.

In this paper, the α-Fe2O3 nanowire arrays were synthesized by a facile hydrothermal process using only FeCl3 as the precursor, which has never been reported before.

  1. In the introduction it is necessary to describe the possible synthesis methods and explain their advantages and disadvantages. At the moment it is not clear if this is the first time when hydrothermal synthesis was applied to obtain Fe2O3.

Response: Thank you very much for your comments and suggestions.

In the introduction, we have described the possible synthesis methods and explain their advantages and disadvantages in the revised manuscript. Also, the hydrothermal process applied in this paper has never been reported before.

Recently, many preparation processes including the magnetron sputtering technique [4, 5], the dip coating method [6, 15, 16, 20], the spin coating technique [7, 9, 14, 21], the ultrasonic spray pyrolysis [8], the hydrothermal method [10, 11, 17, 18], the co-precipitation method [12, 13], and the atomic layer deposition process [19] have been carried out to prepare the Fe2O3 based memory devices. Among them, the hydrothermal method is suitable for the controlled synthesis of the large-scale α-Fe2O3 nanomaterials owing to its low cost, simple process, relatively mild reaction conditions, which is one of the effective methods to prepare the α-Fe2O3 nanowire arrays.

  1. Abstract and conclusions are more or less the same. Authors should overwrite them according to requirements for abstracts and conclusions.

Response: Thank you very much for your comments and suggestions.

The conclusions have been overwritten in the revised manuscript.

In summary, the rhombohedral phase α-Fe2O3 nanowire arrays with the preferential growth orientation along the [110] direction and an average diameter of approximately 37.5 nm have been fabricated by a facile hydrothermal method. The α-Fe2O3 nanowire-based W/α-Fe2O3/FTO memory device has been prepared for the first time. The as prepared α-Fe2O3 nanowire-based W/α-Fe2O3/FTO memory device indicates the nonvolatile bipolar resistive switching behavior with a relatively lower Vset (+0.98 V) and an appropriate resistance ratio of more than two orders of magnitude, which can be preserved for more than 103 s without obvious deterioration. Furthermore, the carrier transport properties of the as prepared W/α-Fe2O3/FTO memory device are associated with the Ohmic conduction mechanism in the low resistance state and the trap-controlled space-charge-limited current conduction mechanism in the high resistance state, respectively. In addition, the partial formation and breakup of conducting nanofilaments modified by the intrinsic oxygen vacancies in the as prepared α-Fe2O3 nanowire arrays have been suggested to be responsible for the nonvolatile bipolar resistive switching behavior of the W/α-Fe2O3/FTO memory device. This work demonstrates that the as prepared α-Fe2O3 nanowire-based W/α-Fe2O3/FTO memory device may be a promising candidate for applications in the future nonvolatile memory devices.

  1. Figure 1 is not described in the text.

Response: Thank you very much for your comments and suggestions.

Figure 1 has been described in the revised manuscript.

The α-Fe2O3 nanowire arrays were synthesized by a facile hydrothermal process, as shown in Figure 1.

  1. Please full file XRD spectra by etalon spectra of No 33-0664.

Response: Thank you very much for your comments and suggestions.

We have given the XRD spectra by etalon spectra of No 33-0664 in the revised manuscript, as shown in Figure 2(a).

Figure 2. (a) XRD patterns of the as prepared α-Fe2O3/FTO, FTO and the standard diffraction peaks of α-Fe2O3 (PDF# 33-0664, JCPDS), respectively.

  1. Probably authors used PDF-2 or PDF-4 database but not JCPDS?

Response: Thank you very much for your comments and suggestions.

We used the PDF-4 database in the revised manuscript.

It is clear that all the XRD diffraction peaks appeared upon the α-Fe2O3 nanowire arrays are indexed to the rhombohedral phase α-Fe2O3 (PDF# 33-0664, JCPDS) [33, 37].

Reviewer 2 Report

In this manuscript, the authors propose the synthesis of nanowire arrays through a hydrothermal method and its resistive switching behavior as potential resistive switching random access memory (ReRAM). Even though the hydrothermal is straightforward, the title is misleading since the device proposed in this manuscript is formed not only of α-Fe2O3 nanowires but is a composite formed by W/α-Fe2O3/FTO, prepared in a two-step synthesis: hydrothermal and magnetron sputtering (fig.1). After careful reading major comments should be addressed:

The introduction describes the W/α-Fe2O3/FTO and α-Fe2O3 materials without differentiation: “In this work, a facile hydrothermal process has been performed to synthesize the hematite α-Fe2O3 nanowire arrays with a preferential growth orientation along the [110] direction on the FTO substrates. The nonvolatile resistive switching behavior of the W/α-Fe2O3/FTO memory device has been achieved,” where two different materials are presented.

The introduction claims: “The partial formation and rupture of conducting nanofilaments modified by oxygen vacancies have been suggested to be responsible for the nonvolatile resistive switching behavior of the W/α-Fe2O3/FTO memory device”. However, there is no characterization of the composite, so there is not possible to make that conclusion.

The XRD spectra have peaks that are not labeled (one at around 25 2θ degrees)

SEM figures 2 b and c must have a scale bar.

The cross-section SEM figure can be improved since there is no clarity from where the 550nm are measured.

Since the nanowire appears to be spear-shaped, with a tip narrower than the bottom, there should be a clear explanation of where the diameter of the nanowire is taken to be 37.5 nm.

There is no characterization of the complete composite, and there is no evidence of tungsten deposition. More detailed characterization should be made with the whole composite.

Determine which peaks are considered to calculate the Scherrer equation. Usually, at least three major peaks are used, but here only two peaks are reported in Fig 2a.

A new fitting of the XPS spectra is strongly suggested, where several issues must be revised. First, for transition metal oxides, it is recommended to use a Gaussian-Lorentzian fitting curve and a Shirley background to start the deconvolution.

The Fe 2p3/2 is usually deconvoluted to a multiplet split spectra, where there is more understanding of the formed species (revise https://doi.org/10.1016/j.apsusc.2010.10.051)

The authors state: “The fitting results of the XPS spectra suggest that a considerable amount of oxygen vacancies exist in the α-Fe2O3 nanowire” However, this statement has no base since there are no calculated values of oxygen vacancies percentage, which are usually made from the areas of the deconvoluted curves. In addition, there is no characterization after the magnetron sputtering, where tungsten is added and could potentially modify the oxygen vacancies.

Format details also should be taken care of: The reagents must have the weight percent of purity, SEM images must have a scale bar, and the Figure 6 resolution must be improved. 

The discussion should be improved, comparing the work with previous studies (references) and not only presenting the results.

General revision of grammar is recommended. 

Author Response

Thank you very much for your letter and for the reviewer’s comments concerning our manuscript entitled “A facile hydrothermal synthesis and resistive switching behavior of α-Fe2O3 nanowire arrays”. (Manuscript ID: molecules-2337904).

Those comments are all valuable and very helpful for revising and improving our paper, as well as the important guiding significance to our researches. The manuscript has been carefully revised and the major revisions were marked in red. The point-by-point answers to the comments and suggestions were listed as below.

Responds to the reviewer’s comments:

In this manuscript, the authors propose the synthesis of nanowire arrays through a hydrothermal method and its resistive switching behavior as potential resistive switching random access memory (ReRAM). Even though the hydrothermal is straightforward, the title is misleading since the device proposed in this manuscript is formed not only of α-Fe2O3 nanowires but is a composite formed by W/α-Fe2O3/FTO, prepared in a two-step synthesis: hydrothermal and magnetron sputtering (fig.1). After careful reading major comments should be addressed:

  1. The introduction describes the W/α-Fe2O3/FTO and α-Fe2O3 materials without differentiation: “In this work, a facile hydrothermal process has been performed to synthesize the hematite α-Fe2O3 nanowire arrays with a preferential growth orientation along the [110] direction on the FTO substrates. The nonvolatile resistive switching behavior of the W/α-Fe2O3/FTO memory device has been achieved,” where two different materials are presented.

Response: Thank you very much for your comments and suggestions.

In this paper, the as prepared α-Fe2O3 nanowire-based W/α-Fe2O3/FTO memory device is composed of the top W electrode, the α-Fe2O3 nanowire arrays and the bottom FTO electrode. The α-Fe2O3 nanowire arrays were synthesized by a facile hydrothermal process, and the circular top W electrodes were directly deposited on the α-Fe2O3 nanowire arrays by the magnetron sputtering process.

  1. The introduction claims: “The partial formation and rupture of conducting nanofilaments modified by oxygen vacancies have been suggested to be responsible for the nonvolatile resistive switching behavior of the W/α-Fe2O3/FTO memory device”. However, there is no characterization of the composite, so there is not possible to make that conclusion.

Response: Thank you very much for your comments and suggestions.

In this paper, the partial formation and rupture of conducting nanofilaments modified by oxygen vacancies are the possible conduction mechanism for the as prepared α-Fe2O3 nanowire-based W/α-Fe2O3/FTO memory device, according to the experimental results as shown in Figure 6 and Figure 7. However, we accept the referee’s suggestion, and we will present the characterization of the composite in the following researches if we have the corresponding test condition.

  1. The XRD spectra have peaks that are not labeled (one at around 25 2θ degrees)

Response: Thank you very much for your comments and suggestions.

The XRD spectra have been revised in the revised manuscript, as shown in Figure 2(a).

Figure 2. (a) XRD patterns of the as prepared α-Fe2O3/FTO, FTO and the standard diffraction peaks of α-Fe2O3 (PDF# 33-0664, JCPDS), respectively.

  1. SEM figures 2 b and c must have a scale bar.

Response: Thank you very much for your comments and suggestions.

The scale bares of the SEM figures 2 b and c have been presented in the revised manuscript, as shown in Figure 2(b) and Figure 2(c).

Figure 2. (a) XRD patterns of the as prepared α-Fe2O3/FTO, FTO and the standard diffraction peaks of α-Fe2O3 (PDF# 33-0664, JCPDS), respectively. (b) Top-view FESEM image of the as prepared α-Fe2O3 nanowire arrays, (c) Cross-sectional FESEM image of the as prepared α-Fe2O3 nanowire arrays. (d) Diameter distribution histogram of the as prepared α-Fe2O3 nanowire arrays.

  1. The cross-section SEM figure can be improved since there is no clarity from where the 550nm are measured.

Response: Thank you very much for your comments and suggestions.

The cross-section SEM figure has been improved in the revised manuscript, as shown in Figure 2(c).

  1. Since the nanowire appears to be spear-shaped, with a tip narrower than the bottom, there should be a clear explanation of where the diameter of the nanowire is taken to be 37.5 nm.

Response: Thank you very much for your comments and suggestions.

The diameter of the α-Fe2O3 nanowire arrays has been evaluated according to the experimental result, as shown in Figure 2(c) in the revised manuscript. Though the α-Fe2O3 nanowire arrays are spear-shaped with a tip narrower than the bottom.

  1. There is no characterization of the complete composite, and there is no evidence of tungsten deposition. More detailed characterization should be made with the whole composite.

Response: Thank you very much for your comments and suggestions.

In this paper, the circular top W electrodes with a diameter of 10 μm were directly deposited on the α-Fe2O3 nanowire arrays by the magnetron sputtering process with a metal shadow mask. The circular top W electrodes are very small for the device, which are difficult to characterize by a normal test. Thus, there is no characterization of the complete composite in this paper. However, we accept the referee’s suggestion, and we will present the characterization of the complete composite in the following researches if we have the corresponding test condition.

  1. Determine which peaks are considered to calculate the Scherrer equation. Usually, at least three major peaks are used, but here only two peaks are reported in Fig 2a.

Response: Thank you very much for your comments and suggestions.

The XRD spectra have been revised in the revised manuscript as shown in Figure 2(a), in which three major peaks have been displayed and been considered to calculate the Scherrer equation.

Figure 2. (a) XRD patterns of the as prepared α-Fe2O3/FTO, FTO and the standard diffraction peaks of α-Fe2O3 (PDF# 33-0664, JCPDS), respectively.

  1. A new fitting of the XPS spectra is strongly suggested, where several issues must be revised. First, for transition metal oxides, it is recommended to use a Gaussian-Lorentzian fitting curve and a Shirley background to start the deconvolution. The Fe 2p3/2 is usually deconvoluted to a multiplet split spectra, where there is more understanding of the formed species (revise https://doi.org/10.1016/j.apsusc.2010.10.051)

Response: Thank you very much for your comments and suggestions.

In general, the fitting results of the XPS spectra only suggest the existence of oxygen vacancies in the α-Fe2O3 nanowire for the α-Fe2O3 nanowire-based W/α-Fe2O3/FTO memory device. However, we accept the referee’s suggestion, and we will use the Gaussian-Lorentzian fitting curve and Shirley background to start the deconvolution in the following researches.

  1. The authors state: “The fitting results of the XPS spectra suggest that a considerable amount of oxygen vacancies exist in the α-Fe2O3 nanowire” However, this statement has no base since there are no calculated values of oxygen vacancies percentage, which are usually made from the areas of the deconvoluted curves. In addition, there is no characterization after the magnetron sputtering, where tungsten is added and could potentially modify the oxygen vacancies.

Response: Thank you very much for your comments and suggestions.

In this paper, the fitting results of the XPS spectra only suggest the presence of oxygen vacancies in the α-Fe2O3 nanowire. However, we accept the referee’s suggestion, and we will present the calculated values of oxygen vacancies percentage in the following researches.

In general, the effect of tungsten on the oxygen vacancies is very small for device. Thus, there is no characterization after the magnetron sputtering in this paper.

  1. Format details also should be taken care of: The reagents must have the weight percent of purity, SEM images must have a scale bar, and the Figure 6 resolution must be improved.

Response: Thank you very much for your comments and suggestions.

The corresponding format details have been revised in the revised manuscript.

  1. The discussion should be improved, comparing the work with previous studies (references) and not only presenting the results.

Response: Thank you very much for your comments and suggestions.

The comparison of the performance parameters for the α-Fe2O3 based memory devices has been tabulated in Table 1 in the revised manuscript.

Table 1. Comparison of the performance parameters of the α-Fe2O3 based memory devices.

Device Structure

Vset/Vreset (V)

Preparation Method

RHRS/RLRS Ratio

Retention

Reference

Ag/Fe2O3/Ti

Pt/Fe2O3/Pt/Ti

Ag/[BiFeO3/γ-Fe2O3]/FTO

Au/Pt-Fe2O3/Ti

Ag/γ-Fe2O3 films/FTO

Au/Fe2O3/FTO

Ag/Fe2O3/ZnO/ITO

~+0.02/−0.1

+1.4/+0.5

+0.98/-1.38

~−1/~+3.4

+1.85/−1.25

~+1.5/~-1.2

+0.9/−1

-

Magnetron sputtering technique

Magnetron sputtering technique

Dip coating method

Spin coating technique

Ultrasonic spray pyrolysis

Spin coating technique

~10

~7

~10

~10

-

~10

~90.1

3.6×103 s

-

-

5×104 s

-

10 hours

30 days

[3]

[4]

[5]

[6]

[7]

[8]

[9]

Ag/Fe2O3/FTO

~+2/~-2

Hydrothermal method

~104

-

[10]

Ag/[TiO2/α-Fe2O3]/FTO

~+4/~−4

Hydrothermal method

~10

103 s

[11]

Ag/Fe2O3-PVA/FTO

+2/~−0.7

Co-precipitation method

~10

-

[12]

Ag/BaTiO3/γ-Fe2O3/ZnO/Ag

top-probe/α-Fe2O3/ZnO/bottom-probe

W/Fe2O3 NC film/Pt

+3.1/-4.7

−0.55/-

~-1.2/~+1.6

Co-precipitation method

Spin coating technique

Dip coating method

~10

~20

>102

-

103 s

105 s

[13]

[14]

[15]

Ti/γ-Fe2O3-NPs/Pt

+1~+2/−1

Dip coating method

~102

-

[16]

Ti/Pt-Fe2O3 core-shell NPs/Pt/PES

Ti/Pt-Fe2O3-core-shell/γ-Fe2O3/Pt

Ti/Fe2O3-SiO2/Si

Cr/ZnO/Pt-Fe2O3 NPs/ZnO/Cr

Au/HfSiO/γ-Fe2O3/Ni2O3/HfSiO/Pt

+2.5/-

~+1/~-1

+2/-2.5

-7/+7

+1.96/-1.90

Hydrothermal method

Hydrothermal method

Atomic layer deposition process

Dip coating method

Spin coating technique

<10

~102

2.8

~5

~102

-

-

-

104 s

-

[17]

[18]

[19]

[20]

[21]

W/α-Fe2O3/FTO

+0.98/−3.11

Hydrothermal method

>102

>103 s

This work

  1. General revision of grammar is recommended.

Response: Thank you very much for your comments and suggestions.

The general revision of grammar has been recommended in the revised manuscript.

Reviewer 3 Report

Minor Revision

The authors systemically investigated A facile hydrothermal process has been developed to synthesize the α-Fe2O3 nanowire

arrays. This work suggests that the W/α-Fe2O3/FTO memory device may be a potential candidate for next-generation nonvolatile memory applications, following issues I am very concern for this work:

1.    How to ensure the uniformity of the film by hydrothermal method? Please provide the device-to-device data of the nanowire arrays.

2.    Why is the top electrode W selected for device? Do other electrodes produce the same effect?

3.    This device read time can be stably preserved for over 103 s without obvious decline. What is the maximum retention time of the device? This device is the promising potential of the memory device for next-generation nonvolatile memory applications. Please provide an experiment related to the memory device.

Author Response

Thank you very much for your letter and for the reviewer’s comments concerning our manuscript entitled “A facile hydrothermal synthesis and resistive switching behavior of α-Fe2O3 nanowire arrays”. (Manuscript ID: molecules-2337904).

Those comments are all valuable and very helpful for revising and improving our paper, as well as the important guiding significance to our researches. The manuscript has been carefully revised and the major revisions were marked in red. The point-by-point answers to the comments and suggestions were listed as below.

Responds to the reviewer’s comments:

The authors systemically investigated A facile hydrothermal process has been developed to synthesize the α-Fe2O3 nanowire arrays. This work suggests that the W/α-Fe2O3/FTO memory device may be a potential candidate for next-generation nonvolatile memory applications, following issues I am very concern for this work:

  1. How to ensure the uniformity of the film by hydrothermal method? Please provide the device-to-device data of the nanowire arrays.

Response: Thank you very much for your comments and suggestions.

The as prepared α-Fe2O3 nanowire-based W/α-Fe2O3/FTO memory device is composed of the top W electrode, the α-Fe2O3 nanowire arrays and the bottom FTO electrode.

As shown in Figure 2(b), the size distribution on the entire surface of the α-Fe2O3 nanowire arrays is relatively uniform. It can be found that the α-Fe2O3 nanowire arrays with a length of about 550 nm have been observed, which can be acted as the dielectric material layer of the W/α-Fe2O3/FTO memory device. In addition, the diameter distribution histogram of the as prepared α-Fe2O3 nanowire arrays further indicates that the average diameter of the α-Fe2O3 nanowire arrays is approximately 37.5 nm, as revealed in Figure 2(d).

The circular top W electrodes with a diameter of 10 μm in the as prepared α-Fe2O3 nanowire-based W/α-Fe2O3/FTO memory device were directly deposited on the α-Fe2O3 nanowire arrays by the magnetron sputtering process with a metal shadow mask.

  1. Why is the top electrode W selected for device? Do other electrodes produce the same effect?

Response: Thank you very much for your comments and suggestions.

The top electrode W has been selected for device because of it is an inert metal electrode with a work function of about 4.6 eV. Generally, it is an Ohmic contact between W and α-Fe2O3 nanowire arrays (because the work function of α-Fe2O3 is about 5.8 eV, which is higher than that of the top electrode W).

In general, the different electrodes with the different work functions would produce the different effects for device.

  1. This device read time can be stably preserved for over 103 s without obvious decline. What is the maximum retention time of the device? This device is the promising potential of the memory device for next-generation nonvolatile memory applications. Please provide an experiment related to the memory device.

Response: Thank you very much for your comments and suggestions.

This device read time can be stably preserved for over 103 s without obvious decline.

In general, we usually test the minimum retention time of the device, and the maximum retention time is usually much larger than 103 s for the device, according to the experimental result as shown in Figure 5(d) in the revised manuscript.

Figure 5. (d) Retention capability of the device.

This device is the promising potential of the memory device for next-generation nonvolatile memory applications. The comparison of the performance parameters for the α-Fe2O3 based memory devices has been tabulated in Table 1 in the revised manuscript.

Table 1. Comparison of the performance parameters of the α-Fe2O3 based memory devices.

Device Structure

Vset/Vreset (V)

Preparation Method

RHRS/RLRS Ratio

Retention

Reference

Ag/Fe2O3/Ti

Pt/Fe2O3/Pt/Ti

Ag/[BiFeO3/γ-Fe2O3]/FTO

Au/Pt-Fe2O3/Ti

Ag/γ-Fe2O3 films/FTO

Au/Fe2O3/FTO

Ag/Fe2O3/ZnO/ITO

~+0.02/−0.1

+1.4/+0.5

+0.98/-1.38

~−1/~+3.4

+1.85/−1.25

~+1.5/~-1.2

+0.9/−1

-

Magnetron sputtering technique

Magnetron sputtering technique

Dip coating method

Spin coating technique

Ultrasonic spray pyrolysis

Spin coating technique

~10

~7

~10

~10

-

~10

~90.1

3.6×103 s

-

-

5×104 s

-

10 hours

30 days

[3]

[4]

[5]

[6]

[7]

[8]

[9]

Ag/Fe2O3/FTO

~+2/~-2

Hydrothermal method

~104

-

[10]

Ag/[TiO2/α-Fe2O3]/FTO

~+4/~−4

Hydrothermal method

~10

103 s

[11]

Ag/Fe2O3-PVA/FTO

+2/~−0.7

Co-precipitation method

~10

-

[12]

Ag/BaTiO3/γ-Fe2O3/ZnO/Ag

top-probe/α-Fe2O3/ZnO/bottom-probe

W/Fe2O3 NC film/Pt

+3.1/-4.7

−0.55/-

~-1.2/~+1.6

Co-precipitation method

Spin coating technique

Dip coating method

~10

~20

>102

-

103 s

105 s

[13]

[14]

[15]

Ti/γ-Fe2O3-NPs/Pt

+1~+2/−1

Dip coating method

~102

-

[16]

Ti/Pt-Fe2O3 core-shell NPs/Pt/PES

Ti/Pt-Fe2O3-core-shell/γ-Fe2O3/Pt

Ti/Fe2O3-SiO2/Si

Cr/ZnO/Pt-Fe2O3 NPs/ZnO/Cr

Au/HfSiO/γ-Fe2O3/Ni2O3/HfSiO/Pt

+2.5/-

~+1/~-1

+2/-2.5

-7/+7

+1.96/-1.90

Hydrothermal method

Hydrothermal method

Atomic layer deposition process

Dip coating method

Spin coating technique

<10

~102

2.8

~5

~102

-

-

-

104 s

-

[17]

[18]

[19]

[20]

[21]

W/α-Fe2O3/FTO

+0.98/−3.11

Hydrothermal method

>102

>103 s

This work

Round 2

Reviewer 1 Report

no comments.

Author Response

Thank you very much for your letter and for the reviewer’s comments concerning our manuscript entitled “A facile hydrothermal synthesis and resistive switching behavior of α-Fe2O3 nanowire arrays”. (Manuscript ID: molecules-2337904).

Those comments are all valuable and very helpful for revising and improving our paper, as well as the important guiding significance to our researches. The manuscript has been carefully revised and the major revisions were marked in red. 

Reviewer 2 Report

1. The title is still misleading since the tungsten is not included as part of the device. 

2. Regarding to this response:

"In this paper, the fitting results of the XPS spectra only suggest the presence of oxygen vacancies in the α-Fe2O3 nanowire. However, we accept the referee’s suggestion, and we will present the calculated values of oxygen vacancies percentage in the following researches. 

In general, the effect of tungsten on the oxygen vacancies is very small for device. Thus, there is no characterization after the magnetron sputtering in this paper." 

Generally, all metal oxides have oxygen vacancies and the XPS spectra show the contribution of the whole sample. If the authors only measured the XPS for the α-Fe2O3 nanowire, there is no comparison with the XPS spectra of the composite, so the affirmation that the XPS spectra only suggest the presence of oxygen vacancies in the α-Fe2O3 nanowire is not valid. Also, with the data available from the deconvolution, it is possible to calculate the percentage of oxygen vacancies, as well as the other oxygen present, therefore it is suggested to include it in this work.

Author Response

Thank you very much for your letter and for the reviewer’s comments concerning our manuscript entitled “A facile hydrothermal synthesis and resistive switching behavior of α-Fe2O3 nanowire arrays”. (Manuscript ID: molecules-2337904).

Those comments are all valuable and very helpful for revising and improving our paper, as well as the important guiding significance to our researches. The manuscript has been carefully revised and the major revisions were marked in red. The point-by-point answers to the comments and suggestions were listed as below.

Responds to the reviewer’s comments:

  1. The title is still misleading since the tungsten is not included as part of the device.

Response: Thank you very much for your comments and suggestions.

We accept the reviewer’s suggestion and the title has been revised as “Synthesis and nonvolatile resistive switching behavior of the α-Fe2O3 nanowire-based W/α-Fe2O3/FTO memory device” in the revised manuscript.

  1. Regarding to this response: "In this paper, the fitting results of the XPS spectra only suggest the presence of oxygen vacancies in the α-Fe2O3 nanowire. However, we accept the referee’s suggestion, and we will present the calculated values of oxygen vacancies percentage in the following researches. In general, the effect of tungsten on the oxygen vacancies is very small for device. Thus, there is no characterization after the magnetron sputtering in this paper."

Generally, all metal oxides have oxygen vacancies and the XPS spectra show the contribution of the whole sample. If the authors only measured the XPS for the α-Fe2O3 nanowire, there is no comparison with the XPS spectra of the composite, so the affirmation that the XPS spectra only suggest the presence of oxygen vacancies in the α-Fe2O3 nanowire is not valid. Also, with the data available from the deconvolution, it is possible to calculate the percentage of oxygen vacancies, as well as the other oxygen present, therefore it is suggested to include it in this work.

Response: Thank you very much for your comments and suggestions.

In the revised manuscript, the survey XPS spectra of the composite have been given as shown in Figure 3(a). It is found that all the Fe and O elements, together with the C element can be observed in the as prepared α-Fe2O3 nanowire arrays, where the C element comes from the carbon source in air adsorbed on the surface of the as prepared α-Fe2O3 nanowire arrays, which is employed to calibrate the other elements including the Fe and O elements.

Also, with the data available from the deconvolution, the O 1s high resolution XPS spectrum can be decomposed into three Gaussian fitting peaks corresponding to 529.53 eV, 530.85 eV, and 532.57 eV, which are assigned to the lattice oxygen, oxygen vacancies and the chemisorbed oxygen species [14, 33, 38], respectively. In particular, the relative concentration of oxygen vacancies can be calculated to be 33.8% from the peak area, which is larger than that of the chemisorbed oxygen species (23.6%) for the as prepared α-Fe2O3 nanowire arrays. The fitting results of the XPS spectra suggest that a considerable amount of oxygen vacancies exist in the as prepared α-Fe2O3 nanowire arrays, which can act as the trapping center and be responsible for the nonvolatile resistive switching behavior of the as prepared α-Fe2O3 nanowire-based W/α-Fe2O3/FTO memory device.

Figure 3. (a) Survey XPS spectra of the as prepared α-Fe2O3 nanowire arrays; (b) Fe 2p, and (c) O 1s high resolution XPS spectra of the as prepared α-Fe2O3 nanowire arrays.

Furthermore, the as prepared α-Fe2O3 nanowire arrays display the excellent optical absorption capacity with an absorption edge of about 645 nm in the visible absorption region, as shown in Figure 4(a).

In addition, the optical band gap of the as prepared α-Fe2O3 nanowire arrays can be found to be 1.92 eV, which is lower than that of the pristine α-Fe2O3 [33, 34, 35] (2.2 eV), further confirming the presence of the intrinsic oxygen vacancies in the α-Fe2O3 nanowire arrays, as depicted in Figure 4(b).

Figure 4. (a) UV-Visible absorption spectra of the α-Fe2O3 nanowire arrays, and (b) Tauc plots of the α-Fe2O3 nanowire arrays.
